# Machine Learning Insights on Driving Behaviour Dynamics among Germany, Belgium, and UK Drivers

Stella Roussou [1,*], Thodoris Garefalakis [1], Eva Michelaraki [1], Tom Brijs [2] and George Yannis [1]

1 Department of Transportation Planning and Engineering, National Technical University of Athens, 5 Iroon Polytechniou Str., 15773 Athens, Greece; tgarefalakis@mail.ntua.gr (T.G.); evamich@mail.ntua.gr (E.M.); geyannis@central.ntua.gr (G.Y.)

2 Transportation Research Institute (IMOB), School of Transportation Sciences, UHasselt–Hasselt University, 3500 Hasselt, Belgium; tom.brijs@uhasselt.be

* Correspondence: s_roussou@mail.ntua.gr

**Abstract:** The i-DREAMS project has a core objective: to establish a comprehensive framework that defines, develops, and validates a context-aware 'Safety Tolerance Zone' (STZ). This zone is crucial for maintaining drivers within safe operational boundaries. The primary focus of this research is to conduct a detailed comparison between two machine learning approaches: long short-term memory networks and shallow neural networks. The goal is to evaluate the safety levels of participants as they engage in natural driving experiences within the i-DREAMS on-road field trials. To accomplish this objective, the study gathered a series of trips from a sample group consisting of 30 German drivers, 43 Belgian drivers, and 26 drivers from the United Kingdom. These trips were then input into the aforementioned machine learning methods to reveal the factors contributing to unsafe driving behaviour across various experiment stages. The results obtained highlight the significant positive impact of i-DREAMS' real-time interventions and post-trip assessments on enhancing driving behaviour. Furthermore, it is worth noting that neural networks demonstrated superior performance compared to other algorithms considered within this research context.

**Keywords:** driving behaviour; road safety; long short-term memory network; neural network; machine learning techniques

## 1. Introduction

Road safety stands as a critical worldwide concern, with an alarming annual toll resulting in around 1.3 million lives lost and numerous injuries due to road crashes [1]. These occurrences are shaped by a variety of elements, including human conduct, road layout, safety attributes of vehicles, environmental circumstances, and socioeconomic differences [1]. Emphasising the crucial contribution of drivers to the happening and intensity of road accidents is essential. A considerable part of these incidents can be linked to driving behaviour, underscoring the pivotal role of drivers in research on traffic safety [2]. Acknowledging the seriousness of this concern, the European Union and the World Health Organization have established ambitious objectives to halve fatal traffic accidents from 2021 to 2030. Emerging technology is anticipated to play a crucial role in realising these advancements in road safety [3].

Road safety is influenced by a range of risk factors, encompassing the driver's condition, environmental factors, and traffic conditions. Despite progress in technology and infrastructure, human error remains a notable contributing element to traffic collisions [4]. Nevertheless, the ongoing advancements in autonomous vehicles hold promise for enhancing road safety by reducing the number of crashes caused by human errors [5]. The integration of autonomous vehicles and intelligent monitoring systems holds considerable potential for reducing the impact of human error and establishing a safer road environment

for all users. Numerous studies have explored the influence of various factors on unsafe driving and have sought to develop models for detecting risky driving behaviour [6,7].

Aligned with the main objective of the i-DREAMS project, this research aims to perform a comprehensive comparative analysis of two distinct machine learning methodologies, namely long short-term memory networks and shallow neural networks. To achieve this goal, the study collected a series of driving trips through the i-DREAMS on-road field trials from a diverse sample group composed of 30 German drivers, 43 Belgian drivers, and 26 drivers from the United Kingdom. By analysing the performance and predictive capabilities of these models in the context of the naturalistic driving experiment, this research endeavours to contribute novel insights into the critical factors influencing road safety and advance the field of intelligent transportation systems.

The i-DREAMS project, funded by the European Commission's Horizon 2020 programme, proposes a framework to address these issues by defining, developing, testing, and establishing a 'Safety Tolerance Zone' (STZ) to promote safe driving behaviour. Within the i-DREAMS project, the STZ has been distinguished into three levels: 'Normal', 'Dangerous', and 'Avoidable Accident', to implement interventions to keep drivers operating within acceptable safety boundaries by continuously monitoring risk factors associated with task complexity (e.g., traffic conditions and weather) and coping capacity (e.g., driver's mental state, driving behaviour, and vehicle status). With regards to the levels, the 'Normal' level implies a low risk of a crash, whereas the 'Dangerous' level indicates an elevated potential but not certainty of a crash. The 'Avoidable Accident' rating indicates a high likelihood of a collision, but it also provides enough time for drivers to act and prevent it. The difference between the 'Dangerous' and 'Avoidable Accident' levels is that the 'Avoidable Accident' level requires an immediate response. In this analysis, the i-DREAMS project provides the infrastructure and context for collecting real-world driving behaviour data from on-road field trials used in this research.

The organisation of the paper is outlined as follows: At the beginning, a detailed introduction about the context and objective of the study is provided. This is followed by an extensive literature review, which attempts to highlight the gaps in the literature that this research attempts to address. The methods utilised and the data collection process are highlighted. Finally, the results of the analyses are presented, along with an important discussion and conclusions.

## 2. Literature Review

In recent years, naturalistic driving studies (NDS) have been widely employed to analyse unsafe driving behaviour [8]. Various factors, including traffic conditions, driver attributes, vehicle characteristics, and environmental elements, influence the risk of driving [9]. Recent research endeavours focus on discerning driving behaviours and classifying them as either risky or safe to enhance road safety [10]. Scholars have employed models to assess unsafe driving behaviour, considering factors like the driver's state [11] and specific driver features such as demographics [12], adopting a more human-centred approach. Additionally, other studies [9,10,13] have introduced models to identify unsafe driving patterns based on factors like speed, time to collision, and time to headway.

Measures based on the vehicle primarily involve assessing driving performance by examining the driver's ability while driving. These measures encompass factors such as vehicle speed, acceleration, steering wheel movement, lane position deviation, gear changes, and other relevant parameters [14,15].

The measures related to the vehicle are classified into three primary categories, consolidating recent findings on their potential correlation with the driver's skill level [16]: (i) driver input to the vehicle (e.g., steering, braking); (ii) vehicle response to driver input (e.g., velocity/acceleration, jerk); and (iii) vehicle state relative to the environment (e.g., headway distance, time to lane change). The first two categories can be directly measured by sensors mounted inside the vehicle, while the third category necessitates information

about the driving environment. These indicators offer the advantage of being real-time, continuous, non-intrusive, and reliable [17].

In-vehicle telematics is a novel technology that has the potential to enhance driving behaviour [18]. Additionally, the ongoing advancements in Intelligent Transportation Systems (ITS) and the growing accessibility of real-time data streams from sensors within vehicles, GPS systems, and mobile devices have created fresh avenues for employing machine learning models in real-time risk forecasting and Advanced Driver Assistance Systems (ADAS). Through constant analysis of sensor data and contextual details, these models can offer timely alerts and notifications to drivers, aid in making safer driving choices, and play a role in averting road crashes.

In the existing literature, various methods and techniques, including artificial neural networks, have been employed to identify risky driving behaviour [18]. For instance, Sohn and Shin (2001) [19] employed a neural network (NN) to classify crash severity based on road type, speed before the crash, and the use of protective devices. Zeng et al. (2016) [20] explored the application of NNs to predict the frequency of crashes with different severity classifications, incorporating a rule extraction methodology to understand the model's sensitivity to varying covariate values. Additionally, Abdelwahab and Abdel-Aty (2002) [21] investigated the use of multi-layer perceptrons and radial basis function NNs to analyse road safety around toll gates.

The most frequently used machine learning techniques in driver behaviour analysis are neural networks (NNs), support vector machines (SVMs), Bayesian learners (BLs), and ensemble learners (ELs). Commonly used neural network architectures encompass feedforward neural networks and recurrent neural networks (RNNs), the latter of which involves incorporating feedback into preceding layers [22,23]. Schmidhuber (2015) [24] provides an overview of NNs with many hidden layers (i.e., deep NNs) and design elements of contest-winning NNs (up to 2014). Furthermore, the LSTM-CNN in the deep learning algorithm is mostly used to identify the abnormal driving behaviour of the driver and can achieve better recognition accuracy [25].

Neural networks stand out as one of the most precise machine learning models for analysing the Driving Events dimension. This dimension has been a focal point of active research when examining driver behaviour through machine learning techniques, with subsequent attention given to the physiological and psychological states. The evaluation of the Driving Events dimension indicates that machine learning models exhibit arithmetic means of accuracy ranging approximately from 73% to 98%, recall from 82% to 96%, and specificity from 84% to 97% [26]. Despite numerous research endeavours on analysing driver behaviour through machine learning algorithms, there are currently no comparable studies in this specific domain that investigate both machine learning (ML) and deep learning (DL) algorithms [27]. Therefore, in this context, it is imperative to conduct experimentation and compare the neural network and long short-term memory algorithms. This is essential to elevating and advancing the initial idea into a forward-looking, future-proof framework for the next generation.

Machine learning (ML) and deep learning (DL) models serve as potent instruments for understanding, predicting, and alleviating risky driving behaviour. Their algorithms, combined with vast datasets, possess transformative potential for initiatives aimed at enhancing road safety. Considering the context of this paper, delving into the utilisation and efficacy of neural networks (NNs) and long short-term memory (LSTM) models amidst these challenges and opportunities can offer a nuanced insight into their influence on driving behaviours across diverse cultural contexts. Sustained research and collaboration in this domain are essential to fully leveraging the advantages of sophisticated algorithms in enhancing driving safety, especially in the distinct driving environments of Germany, Belgium, and the United Kingdom.

## 3. Materials and Methods

### 3.1. Data Collection

The driving behaviour data utilised in this analysis originates from the i-DREAMS project's on-road field trials. Within the i-DREAMS project, a naturalistic driving experiment was carried out involving 30 drivers from Germany, and a large database of 5344 trips and 84,434 min was created. As for the Belgian drivers, the database consisted of 43 drivers, 7163 trips, and 147,337 min. Lastly, for the UK car drivers, the dataset included 26 drivers, 8226 trips, and 118,175 min. As shown in Table 1, the on-road trial experiment was carried out in four phases.

**Table 1.** Phases of the on-road experiment.

| Phase 1 | | |
|---|---|---|
| **Country** | **Drivers** | **Trips (Minutes)** |
| Germany | 30 | 1397 trips (23,617 min) |
| Belgium | 39 | 1173 trips (23,725 min) |
| UK | 25 | 618 trips (10,803 min) |
| **Phase 2** | | |
| **Country** | **Drivers** | **Trips (Minutes)** |
| Germany | 30 | 1322 trips (19,469 min) |
| Belgium | 43 | 1549 trips (31,414 min) |
| UK | 26 | 2243 trips (25,151 min) |
| **Phase 3** | | |
| **Country** | **Drivers** | **Trips (Minutes)** |
| Germany | 30 | 1129 trips (17,704 min) |
| Belgium | 51 | 1973 trips (40,121 min) |
| UK | 26 | 2198 trips (24,569 min) |
| **Phase 4** | | |
| **Country** | **Drivers** | **Trips (Minutes)** |
| Germany | 30 | 1496 trips (23,644 min) |
| Belgium | 49 | 2468 trips (52,077 min) |
| UK | 26 | 3167 trips (57,652 min) |

The on-road experiment was conducted following established principles derived from the pertinent literature, with a specific focus on evaluating interventions designed to enhance drivers' adherence to safe driving practices. The experiment encompassed four distinct phases. Phase 1, designated as the monitoring phase, spanned 4 weeks and involved no interventions. Phase 2, spanning 4 weeks as well, introduced in-vehicle interventions by delivering real-time warnings through adaptive Advanced Driver Assistance Systems (ADAS). In Phase 3, also lasting 4 weeks, drivers received feedback on their driving performance through a mobile application. In Phase 4, a 6-week period, drivers continued to receive feedback as in Phase 3, but with the added incorporation of gamification elements. All four phases were focused on the observation of driving behaviour and the assessment of the impact of real-time interventions, including in-vehicle warnings, as well as post-trip interventions like feedback and gamification, on driving behaviour.

### 3.2. Neural Networks (NNs)

Artificial neural networks (ANNs) represent a powerful computational model capable of capturing complex non-linear patterns within datasets. These networks emulate the parallel processing of human neurons and are commonly employed in classification tasks. The architecture used in this study, known as the multi-layer perceptron ANN, is composed of three essential layers: an input layer, one or more hidden layers, and an output layer.

In the context of analysing risky driving behaviour, the input layer functions as the initial data receiver, encompassing various driving attributes like vehicle speed, acceleration, and headway.

The hidden layer, featuring a variable number of neurons, conducts computations by combining weighted inputs from these attributes. Each neuron in the hidden layer is equipped with an activation function, introducing the necessary non-linearity to the model. This non-linearity is crucial for capturing intricate patterns and relationships between these attributes and the target variable, which, in this study, pertains to different levels of risky driving behaviour. The determination of the number of neurons in the hidden layer often involves experimentation, as it significantly impacts the network's capacity to learn and generalise. Simple problems may require just one hidden layer, whereas more complex tasks might demand multiple hidden layers.

Moving to the output layer, it serves as the central hub for consolidating information from the hidden neurons to generate the network's final output. In the context of this classification task regarding risky driving behaviour, the output layer includes multiple neurons, each corresponding to distinct classes or levels of risk. The choice of activation function within the output layer depends on the specific problem. For multi-class classification, the softmax activation function is commonly used to calculate class probabilities, aiding in the prediction of risky driving behaviour levels.

The design and architecture of the neural network, encompassing factors like the number of layers, neurons, and activation functions, play a pivotal role in achieving accurate and effective classification for risky driving behaviour. Previous research in this domain has extensively explored the benefits of multi-layer perceptron ANNs, highlighting their ability to uncover complex patterns and associations hidden within driving data [28,29].

### 3.3. Long Short-Term Memory (LSTM) Networks

A long short-term memory (LSTM) network is a type of recurrent neural network (RNN) that has emerged as a pivotal tool in various fields due to its exceptional ability to capture and model complex sequential patterns. LSTM networks were introduced as a solution to the vanishing gradient problem that commonly affects traditional RNNs, reducing their performance in tasks demanding long-term dependencies [30]. Their intrinsic capability to preserve data over extended periods renders them highly suitable for tasks that require modelling sequential data.

An LSTM network is constructed from a succession of repeating modules, creating a chain-like architectural structure [31]. The fundamental information processing components within LSTM networks are referred to as 'cells', akin to the more intricate counterparts of neurons in traditional multi-layer perceptrons (MLP). Within each LSTM cell, there exist multiple gates, which serve the crucial function of controlling and managing the information flow across sequences of arbitrary length. This intrinsic characteristic empowers LSTM networks to autonomously discern the relevance of information over both long-term and short-term contexts, rendering them a well-suited choice for a wide array of sequential data-related tasks such as activity recognition and language translation [32].

In a typical LSTM configuration, there are three key gates:

1. The Forget Gate: Responsible for determining which information should be retained and which should be forgotten in the cell state. This gate employs a sigmoid layer, known as the "forget gate layer", to make these determinations.
2. The Input Gate: This gate decides what new information should be included in the cell state and how it should be updated. It is composed of two essential components: an input gate layer, utilising a sigmoid function to determine what values should be updated, and a hyperbolic tangent (tanh) layer that produces a vector of candidate values for potential integration into the state. Subsequently, the old cell state undergoes an update based on these components.
3. The Output Gate: Responsible for filtering and selecting the information to be output from the memory block at a particular time step. The output is derived from the

cell state but undergoes filtering. An output gate, which consists of a sigmoid layer, determines the relevant portions of the cell state to be included in the output. The filtered cell state then passes through a tanh activation function to scale values within the range of $-1$ and 1. Finally, the result is multiplied by the output of the sigmoid gate, generating the desired output.

### 3.4. Performance Metrics

For the assessment of classification models, the evaluation of model performance involved the utilisation of the confusion matrix in conjunction with various well-established performance metrics defined by Equations (1)–(5).

Accuracy, which measures the proportion of correctly classified observations, is defined as:

$$\text{Accuracy} = \frac{\text{TP} + \text{TN}}{\text{TP} + \text{FP} + \text{FN} + \text{TN}} \tag{1}$$

Precision, which quantifies the number of positive class predictions that actually belong to the positive class, is defined as follows:

$$\text{Precision} = \frac{\text{TP}}{\text{TP} + \text{FP}} \tag{2}$$

Recall, also known as true positive rate, which measures the proportion of actual positive cases correctly identified by the model, is defined as follows:

$$\text{Recall} = \frac{\text{TP}}{\text{TP} + \text{FN}} \tag{3}$$

The F1-score, which combines precision and recall into a single measure, is defined as follows:

$$\text{F1-score} = \frac{2\text{x} \, (\text{Precision}) * (\text{Recall})}{(\text{Precision}) + (\text{Recall})} \tag{4}$$

The false alarm rate, which measures the proportion of negative cases incorrectly classified as positive, is defined as follows:

$$\text{False Alarm Rate} = \frac{\text{FP}}{\text{FP} + \text{TN}} \tag{5}$$

where true positive (TP) denotes instances belonging to class i that were accurately classified as such. True negative (TN) refers to instances not belonging to class i and correctly not classified as such. False positive (FP) indicates instances that do not belong to class i but were erroneously classified as part of it. Lastly, false negative (FN) signifies instances that belong to class i but were mistakenly not classified as such.

### 3.5. Methodology Analysis

The neural network model is structured as a multi-layer architecture, where each layer is responsible for extracting and learning different levels of patterns from the input driving data. The initial layer processes basic features, which are then passed on to subsequent layers for more complex pattern recognition. The training process involves adjusting the model weights based on historical driving data, enhancing its ability to differentiate between safe and risky driving behaviours. This model's performance is later validated using a separate set of data to determine its predictive accuracy. The following high-level description demonstrated in Figure 1 below presents the methodology structure.

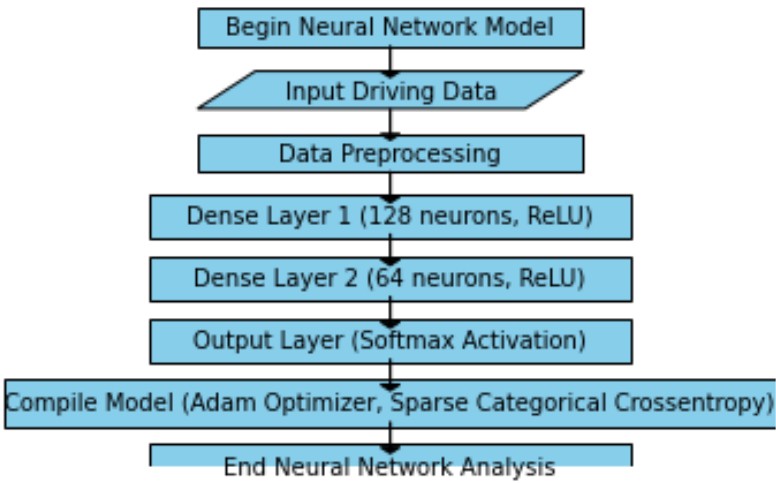

**Figure 1.** High-level algorithm description of the neural network model.

The LSTM model was structured upon the same methodology, as presented below in Figure 2.

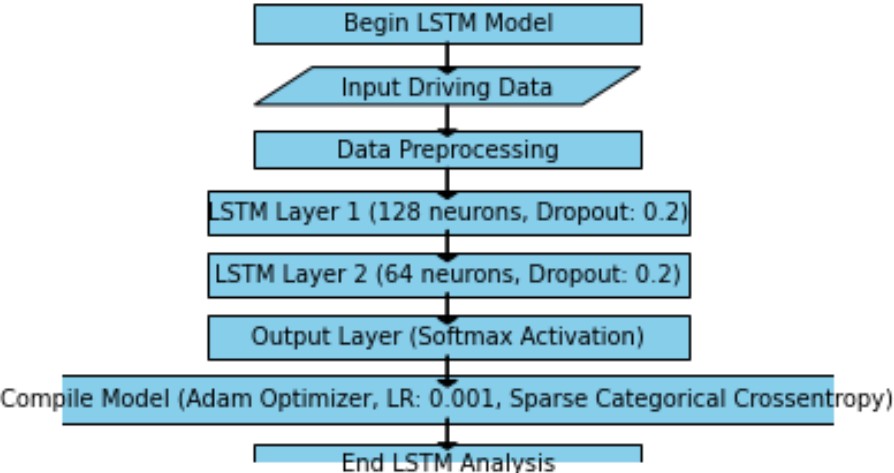

**Figure 2.** High-level algorithm description of the long short-term memory model.

A snippet of the code for the neural network model and long short-term memory model has been provided in Appendix A. It is important to add that the NN model architecture consists of two dense (fully connected) layers. The first layer has 128 units, and the second has 64 units, all using the ReLU activation function. The output layer is configured with the appropriate number of units based on the classes in the target variable, utilising the softmax activation function. The model is optimised using the Adam optimizer with a sparse categorical cross-entropy loss function. During training, the model is optimised for 100 epochs with a batch size of 32, and 10% of the training data are reserved for validation.

The hyperparameter values in the LSTM model were chosen based on established practices for working with similar datasets. The LSTM model is designed for sequential data and is composed of two LSTM layers with 128 and 64 units, respectively. Both LSTM layers use ReLU activation. The ReLU activation function was chosen for its ability to capture complex relationships. A dropout rate of 0.2 and a recurrent dropout of 0.2 were implemented to prevent overfitting. The output layer, employing softmax activation, is adapted based on the number of classes. The model is compiled with the Adam optimizer, a learning rate of 0.001, and sparse categorical cross-entropy loss. Training occurs over 100 epochs with a batch size of 64, and 10% of the training data are set aside for validation.

## 4. Results

*4.1. Neural Networks (NNs) for Headway and Speeding at a Normal Level*

4.1.1. German Car Drivers

In this study, neural network (NN) classification algorithms were applied as a preliminary step for subsequent LSTM classification. Two feed-forward multi-layer perceptron models were utilised on a subset of data from 30 German car drivers and 5340 trips. These models demonstrated exceptional accuracy, exceeding 94%, indicating their effectiveness in real-time prediction of the STZ. This result supports the feasibility of real-time STZ prediction. Additionally, the models exhibited a low false alarm rate, maxing out at 6%, showcasing their ability to minimise incorrect predictions and unnecessary alerts.

Following the application of these models, a confusion matrix was generated for the independent variables of interest, namely headway and speeding, as presented in Table 2 and the performance of neural network classification on headway and speeding STZ level for the German car drivers is presented in Figure 3. This matrix provides valuable insights into the classification's performance and forms the basis for further analysis and discussion in this study.

**Table 2.** Confusion data matrix for headway and speeding of neural network model of German car drivers.

| Variable | TP | FP | FN | TN | Sum |
|---|---|---|---|---|---|
| Headway | 33,378 | 0 | 1400 | 82 | 34,860 |
| Speeding | 2178 | 1987 | 63 | 30,632 | 34,860 |

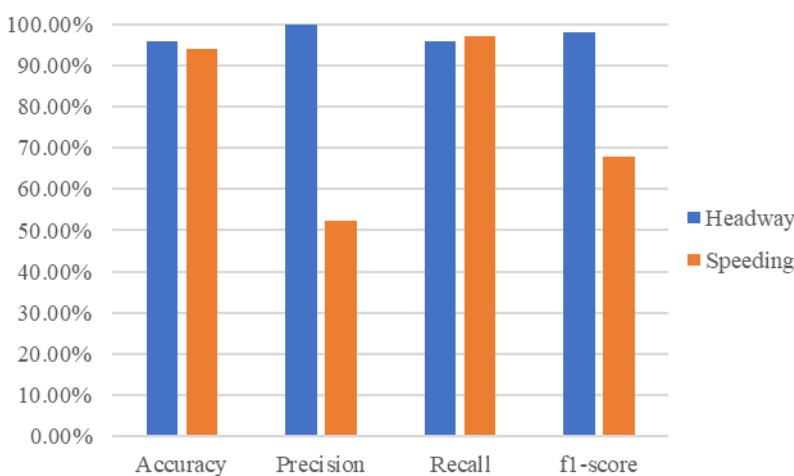

**Figure 3.** Performance of the neural network classification for headway and speeding of neural network model of German car drivers.

From the confusion matrix, the following metrics were estimated and are depicted in Table 3.

**Table 3.** Assessment of the classification model for headway and speeding of neural network model of German car drivers.

| Variable | Accuracy | Precision | Recall | F1-Score |
|---|---|---|---|---|
| Headway | 95.98% | 100.00% | 95.97% | 97.95% |
| Speeding | 94.12% | 52.29% | 97.19% | 68.00% |

In Figure 4, the plot titled "Model Loss" displays the progression of the model's loss during training and validation phases across multiple epochs. The *x*-axis represents the number of training epochs, while the *y*-axis represents the corresponding loss values. The blue line represents the model's training loss at each epoch. Training loss measures how well the model is performing on the training data. As the model learns from the training data, the goal is to minimise this loss, indicating improved predictive performance. The orange line represents the validation loss at each epoch. Validation loss measures how well the model generalises to unseen data not used during training. It helps to identify if the model is overfitting (performing well on training data but poorly on new data) or underfitting (not capturing the underlying patterns).

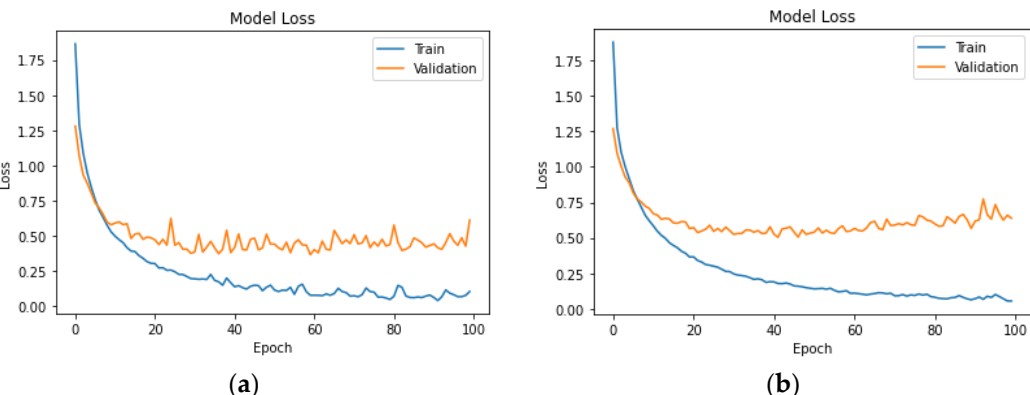

**Figure 4.** Model loss of the neural network of German car drivers for headway (**a**) and speeding (**b**).

A descending trend in both training and validation loss is generally desired and achieved in Figure 4, Figure 6, Figure 8, Figure 10, Figure 12, and Figure 14 of both neural network and long short-term memory models for all three countries for headway (a) and speeding (b). This signifies that the model is learning and improving its predictive ability over epochs. Both training and validation losses converge to a low value, suggesting that the model is effectively learning from the data and generalising well. Divergence or a significant gap between training and validation loss might indicate overfitting (high training performance but poor generalisation) or underfitting (the model is too simple to capture the data patterns).

The outcomes presented align closely with pertinent studies in real-time safety assessments [33], as well as prior analyses conducted on simulator data in similar projects. It is noteworthy that the lower precision and F1-score metrics can be attributed to the larger number of instances depicting 'normal' STZ levels in comparison to instances representing 'dangerous' conditions. This prevalence of 'normal' instances naturally impacts the precision and F1-score values, highlighting the challenges in accurately classifying situations with lower occurrence rates. These nuances emphasise the importance of context and distribution in interpreting classification performance metrics.

It is important to note that model performance can vary across different countries due to distinct driving behaviours, road conditions, and traffic regulations. The F1-score, being a harmonic mean of precision and recall, provides a balanced measure of a model's accuracy, considering both false positives and false negatives. In the context of different countries, variations in driving behaviours and environmental factors may impact the model's performance.

4.1.2. Belgian Car Drivers

The results obtained from the Belgian car drivers' dataset, as illustrated in Tables 4 and 5, demonstrate the models' strong performance. Particularly in the case of speeding prediction, the models achieved remarkable accuracy and recall. For headway prediction, although the accuracy is marginally lower than that of speeding, the balanced precision

and recall values indicate the models' ability to accurately identify true positive cases while minimising false positives and false negatives.

**Table 4.** Confusion data matrix for headway and speeding of neural network model of Belgium car drivers.

| Variable | TP | FP | FN | TN | Sum |
|---|---|---|---|---|---|
| Headway | 37,517 | 0 | 80 | 7915 | 45,512 |
| Speeding | 30,069 | 0 | 0 | 6193 | 36,462 |

**Table 5.** Assessment of the classification model for headway and speeding of neural network model of Belgium car drivers.

| Variable | Accuracy | Precision | Recall | F1-Score |
|---|---|---|---|---|
| Headway | 77.19% | 77.64% | 77.19% | 76.90% |
| Speeding | 83.51% | 80.71% | 83.51% | 79.78% |

These findings strongly suggest the models' effectiveness in classifying instances related to headway and speeding. The speeding model, in particular, exhibited exceptional performance in detecting positive cases. It is noteworthy that both cases showed an absence of false positives (FP = 0), a significant accomplishment indicating a minimal rate of incorrectly identified positive cases. Additionally, a performance of neural network classification on headway and speeding STZ level for the Belgium car drivers is presented in Figure 5.

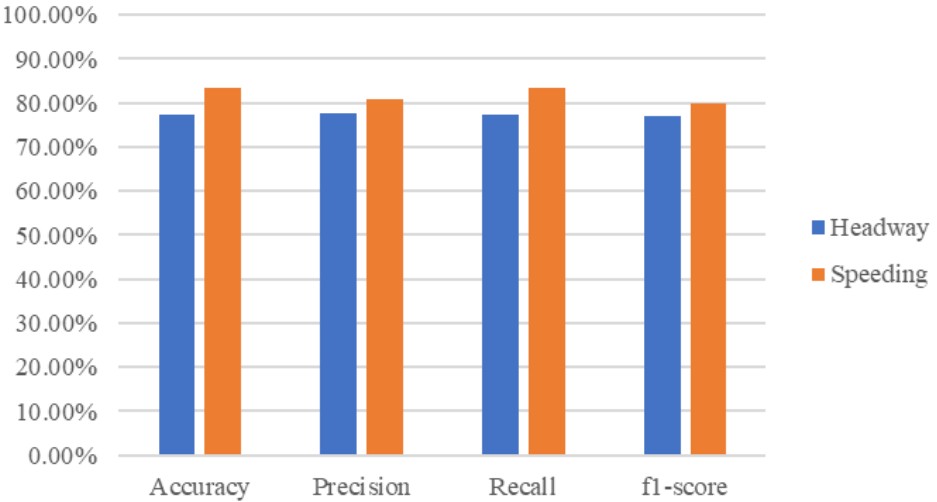

**Figure 5.** Performance of the neural network classification for headway and speeding.

From the confusion matrix, the following metrics were estimated and are depicted in Table 5.

A descending trend in both training and validation loss is achieved in the neural network of Belgium car drivers for headway (a) and speeding (b) as demonstrated in Figure 6 below.

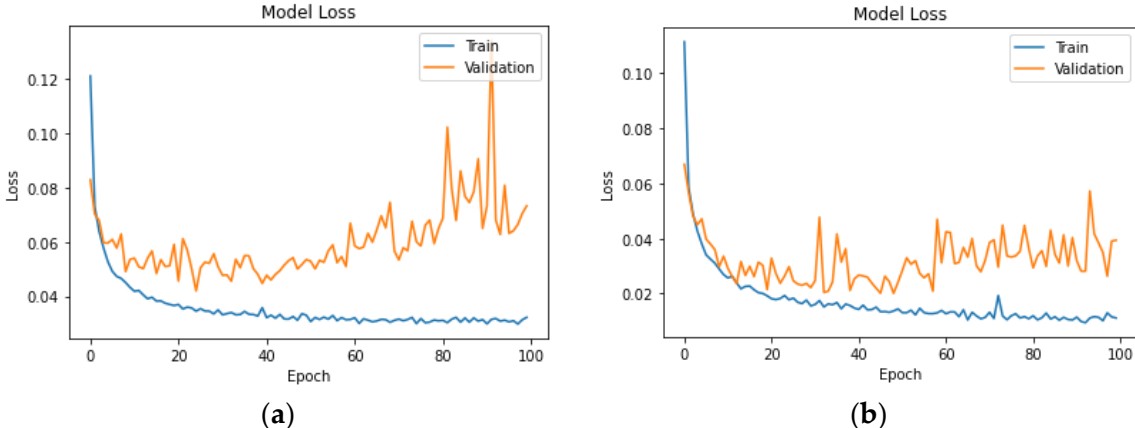

**Figure 6.** Model loss of the neural network of Belgian car drivers for headway (**a**) and speeding (**b**).

Upon analysing the results, it is evident that the models performed exceptionally well, particularly in forecasting instances of speeding. The model not only showcased impressive accuracy, reflecting the overall correctness of its predictions, but also demonstrated a high recall rate. This heightened recall indicates the model's successful identification of the majority of actual positive cases of speeding. Likewise, in headway prediction, although the accuracy was slightly lower than that of speeding, the model maintained a delicate balance between precision and recall. This equilibrium signifies the model's ability to accurately identify true positive cases without generating an excessive number of false positives or missing actual positive instances.

These results underscore the models' robustness and effectiveness in accurately classifying instances of speeding and headway, emphasising their potential for practical applications in real-world scenarios, particularly in the context of enhancing road safety and traffic management.

### 4.1.3. UK Car Drivers

The examination of the results reveals a commendable performance by the models, particularly in the prediction of instances related to headway and speeding. The model exhibited a high level of accuracy, indicating the overall correctness of its predictions in both scenarios. Furthermore, the model demonstrated an impressive recall rate, especially in the case of headway prediction, where it accurately identified a substantial majority of the actual positive cases, balancing precision effectively. Similarly, in the context of speeding prediction, the model maintained a delicate equilibrium between precision and recall, ensuring accurate identification of true positive cases without generating an excessive number of false positives or missing actual positive instances.

These findings, as presented in Tables 6 and 7, highlight the models' reliability and effectiveness in classifying instances of headway and speeding. These results not only showcase their potential for real-world applications but also emphasise their significance in the realm of road safety and traffic management. The absence of false positives in both cases (FP = 0) is a notable achievement, signifying a minimal rate of incorrectly identified positive cases. This reinforces the models' utility in making precise predictions, thereby contributing significantly to the enhancement of road safety standards.

**Table 6.** Confusion data matrix for headway and speeding of neural network model of UK car drivers.

| Variable | TP | FP | FN | TN | Sum |
|---|---|---|---|---|---|
| Headway | 33,617 | 0 | 0 | 6510 | 40,063 |
| Speeding | 21,149 | 0 | 1 | 10,854 | 32,003 |

**Table 7.** Assessment of the classification model for headway and speeding of neural network model of UK car drivers.

| Variable | Accuracy | Precision | Recall | F1-Score |
|----------|----------|-----------|--------|----------|
| Headway | 80.98% | 81.37% | 80.98% | 80.83% |
| Speeding | 79.89% | 79.13% | 79.89% | 74.95% |

From the confusion matrix, the following metrics were estimated and are depicted in Table 7.

A performance of neural network classification on headway and speeding STZ level for the UK car drivers is presented in Figure 7.

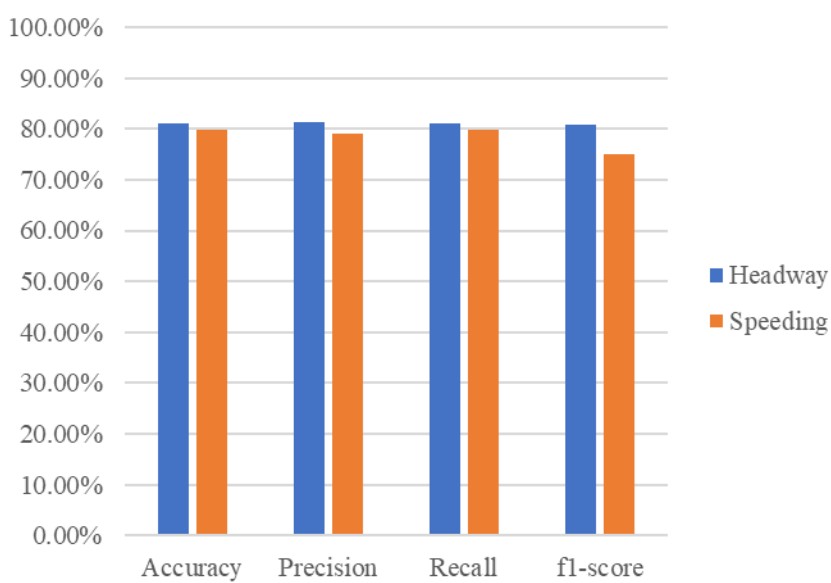

**Figure 7.** Performance of the neural network classification for headway and speeding.

A descending trend in both training and validation loss is achieved in the neural network of UK car drivers for headway (a) and speeding (b) as demonstrated in Figure 8 below.

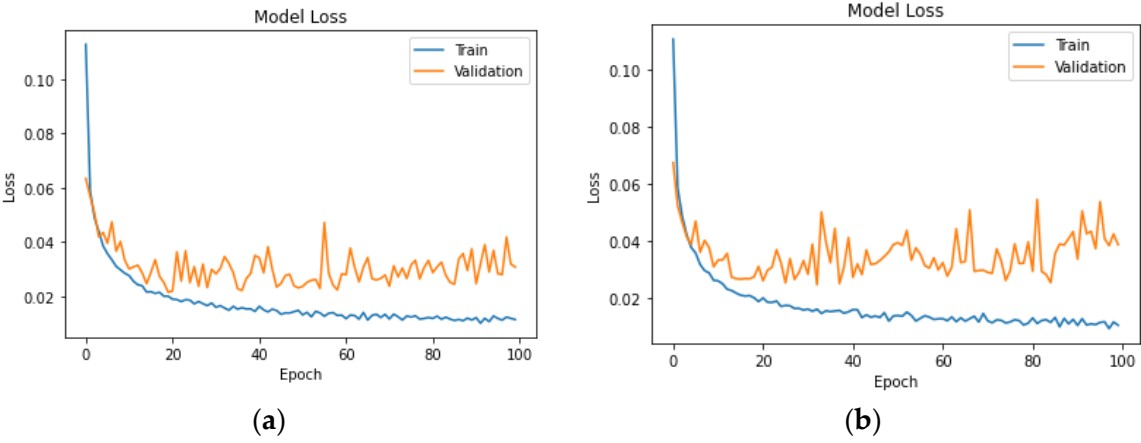

**Figure 8.** Model loss of the neural network for UK car drivers for headway (**a**) and speeding (**b**).

The analysis of headway and speeding prediction models across different countries, namely Germany, Belgium, and the UK, provides valuable insights into their effectiveness in real-world driving scenarios. The German drivers' model exhibited exceptional accuracy, precision, and recall for headway prediction, indicating its robust ability to accurately identify safe following distances. However, in the case of speeding prediction, while the accuracy was high, the model struggled with precision, resulting in a notable number of false positives. This suggests the need for fine-tuning to reduce incorrect identifications of speeding.

In Belgium, the models demonstrated a balanced performance for both headway and speeding predictions. The accuracy and recall rates were consistent, signifying the models' ability to maintain accuracy while identifying a significant portion of positive cases. Precision values, especially in the context of speeding, demonstrated commendable outcomes, signifying an effective balance between accurate positive identifications and minimising false positives. For UK drivers, the headway prediction model showcased strong accuracy, precision, recall, and F1-score, indicating its effectiveness in recognising safe following distances. The speeding prediction model, while maintaining relatively good accuracy and precision, struggled with recall, indicating a challenge in identifying all positive speeding cases. Fine-tuning efforts could enhance the model's sensitivity in detecting speeding instances.

The successful application of these neural network models paves the way for the implementation of LSTM-based approaches, which can leverage the temporal nature of the data to potentially enhance the precision and reliability of the STZ prediction. The upcoming subsection will delve further into the LSTM classification, building upon the foundations established by the neural network models.

*4.2. Long Short-Term Memory (LSTM) for Headway and Speeding*

4.2.1. German Car Drivers

Building upon the foundations laid by the previously mentioned neural network models, the subsequent subsection focuses on the application of long short-term memory (LSTM) classification for real-time prediction of the STZ. The LSTM approach capitalises on the temporal nature of the data to potentially enhance the precision and reliability of the STZ prediction. The LSTM models were trained and evaluated using a subset of the German car drivers' dataset, consisting of data from 30 drivers and 5340 trips.

The LSTM models, while showing a lower level of accuracy and precision compared to the previous neural network models, still exhibit a fair level of performance in predicting headway and speeding incidents.

For headway prediction, as presented in Table 8, the model accurately identifies approximately 58.17% of instances, which is a significant improvement from random chance. The precision of 42.44% indicates that when the model predicts a positive case, it is correct 42.44% of the time. The recall of 58.17% implies that the model captures 58.17% of all actual positive cases. The F1-score of 43.86% signifies a balanced measure of precision and recall.

**Table 8.** Assessment of the classification model for headway and speeding of LSTM of German car drivers.

| Variable | Accuracy | Precision | Recall | F1-Score |
|----------|----------|-----------|--------|----------|
| Headway | 58.17% | 42.44% | 58.17% | 43.86% |
| Speeding | 73.50% | 54.03% | 73.50% | 62.28% |

In the case of predicting speeding, the model performs better with an accuracy of 73.50%. The precision of 54.03% indicates that nearly half of the positive predictions made by the model are accurate. The recall of 73.50% shows that the model captures 73.50% of all actual speeding cases. The F1-score of 62.28% indicates a balanced trade-off between precision and recall.

Compared to the previous neural network models, these LSTM models show a lower level of accuracy and precision. However, it is crucial to note that LSTM models are particularly valuable in capturing sequential patterns and temporal dependencies in data. Despite the decrease in accuracy and precision, the LSTM models might excel at capturing nuanced patterns in the data, especially temporal ones.

Additionally, a performance of LSTM on headway and speeding STZ level for the German car drivers is presented in Figure 9.

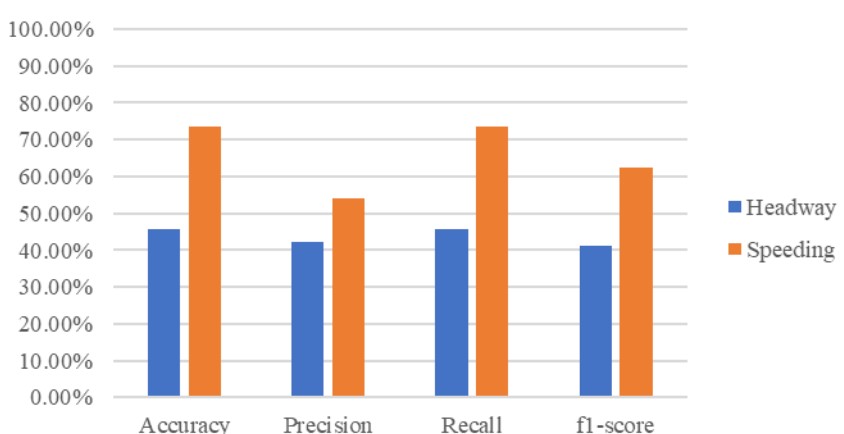

**Figure 9.** Performance of the LSTM model for headway and speeding at a normal level of German car drivers.

A descending trend in both training and validation loss is achieved in the LSTM of German car drivers for headway (a) and speeding (b) as demonstrated in Figure 10 below. The Divergence or this significant gap between training and validation loss in Figure 10a might indicate overfitting (high training performance but poor generalisation).

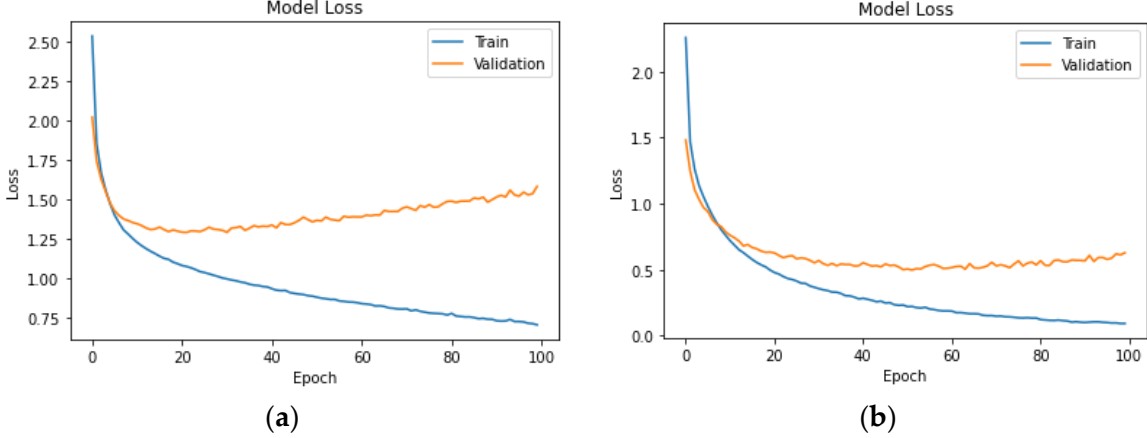

**Figure 10.** Model loss of the LSTM model for German car drivers for headway (**a**) and speeding (**b**).

4.2.2. Belgian Car Drivers

It is important to consider that an accuracy below 60% may not be satisfactory for a high-performance intervention system, as it could result in a relatively high number of false alarms or missed detections. However, the required level of accuracy depends on the specific use case and the associated risks. For instance, in a system aimed at detecting potential crashes or safety hazards, a higher level of accuracy may be necessary to ensure the safety of drivers and other road users.

The LSTM models for Belgium show moderate performance in predicting headway and speeding incidents, as presented in Table 9 below. For headway prediction, the model achieves an accuracy of 58.12%, indicating it correctly classifies approximately 58.12% of the instances. The precision of 35.65% suggests that when the model predicts a positive case, it is correct 35.65% of the time. The recall of 58.12% signifies that the model captures 58.12% of all actual positive headway cases. The F1-score of 37.33% reflects a balance between precision and recall.

**Table 9.** Assessment of the classification model for headway and speeding of LSTM of Belgium car drivers.

| Variable | Accuracy | Precision | Recall | F1-Score |
|----------|----------|-----------|--------|----------|
| Headway | 58.12% | 35.65% | 58.12% | 37.33% |
| Speeding | 48.27% | 25.75% | 48.27% | 32.59% |

In the case of predicting speeding, the model performs slightly lower with an accuracy of 48.27%. The precision of 25.75% indicates that only a quarter of the positive predictions made by the model are accurate. The recall of 48.27% shows that the model captures 48.27% of all actual speeding cases. The F1-score of 32.59% indicates a trade-off between precision and recall.

The LSTM models in Belgium exhibit moderate performance, especially in identifying headway incidents. While they demonstrate a capacity to capture positive cases, there is room for improvement, particularly in reducing false positives and enhancing precision. Further refinements in model architecture, feature selection, or additional data pre-processing techniques might be necessary to enhance the accuracy and reliability of the LSTM models for both headway and speeding predictions.

A performance of LSTM on headway and speeding STZ level for the Belgium car drivers is presented in Figure 11.

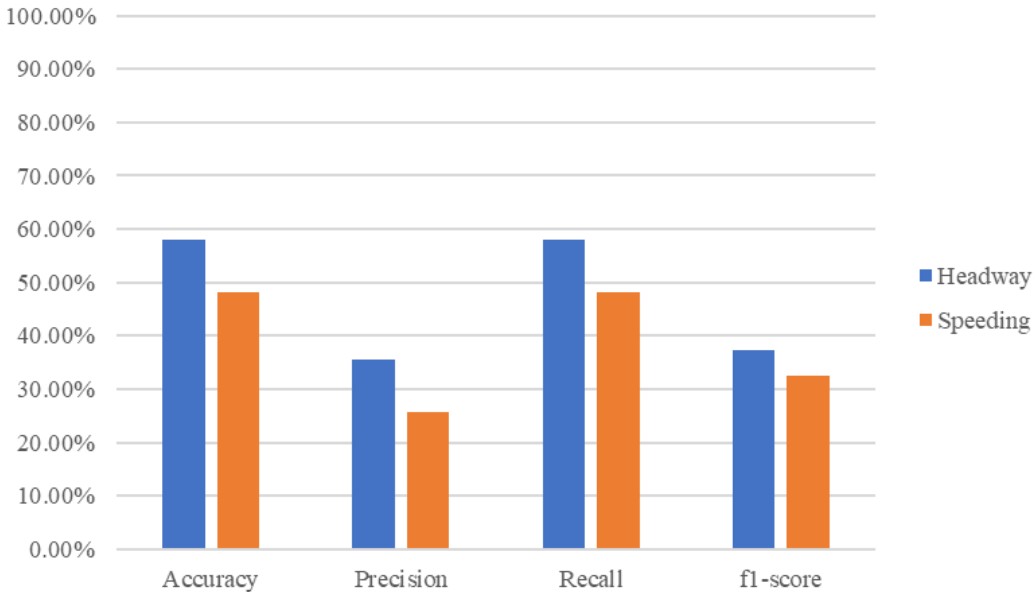

**Figure 11.** Performance of the LSTM model for headway and speeding at a normal level.

A descending trend in both training and validation loss is achieved in the LSTM of Belgium car drivers for headway (a) and speeding (b) as demonstrated in Figure 12 below.

The Divergence or this significant gap between training and validation loss in Figure 12b might indicate overfitting (high training performance but poor generalisation).

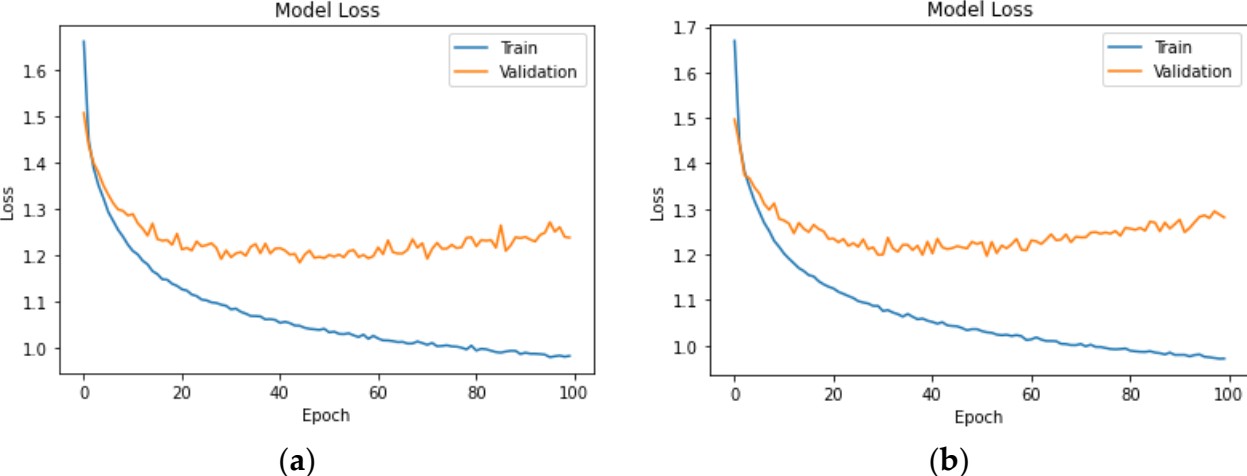

**Figure 12.** Model loss of the LSTM model for Belgian car drivers for headway (**a**) and speeding (**b**).

### 4.2.3. UK Car Drivers

It is crucial to acknowledge that an accuracy level falling below 60% might not meet the requirements for a high-performance intervention system. Such low accuracy can lead to a considerable number of false alarms or missed detections. However, the necessary accuracy threshold depends on the specific use case and associated risks. For instance, in systems designed to detect potential crashes or safety hazards, a higher accuracy level becomes imperative to ensure the safety of drivers and other road users.

Looking specifically at the LSTM models implemented in the UK, their performance in predicting headway and speeding incidents appears to be moderate, as showcased in Table 10. For headway prediction, the model achieves an accuracy of 54.45%, correctly classifying approximately 54.45% of the instances. The precision, indicating the accuracy of positive predictions, stands at 33.72%. This suggests that the model accurately predicts positive cases only 33.72% of the time. Additionally, the recall rate is 54.45%, indicating that the model captures 54.45% of all actual positive headway cases. The F1-score, reflecting a balance between precision and recall, is 40.20%.

**Table 10.** Assessment of the classification model for headway and speeding of LSTM of UK car drivers.

| Variable | Accuracy | Precision | Recall | F1-Score |
|----------|----------|-----------|--------|----------|
| Headway | 54.45% | 33.72% | 54.45% | 40.20% |
| Speeding | 49.51% | 29.58% | 49.51.% | 35.22.% |

In the case of predicting speeding, the model performs slightly lower with an accuracy of 49.51%. The precision, indicating accurate positive predictions, is at 29.58%, implying that only a quarter of the positive predictions made by the model are accurate. The recall rate is 49.51%, suggesting that the model captures 49.51% of all actual speeding cases. The F1-score, representing the trade-off between precision and recall, is 35.22%.

A performance of LSTM on headway and speeding STZ level for the UK car drivers is presented in Figure 13.

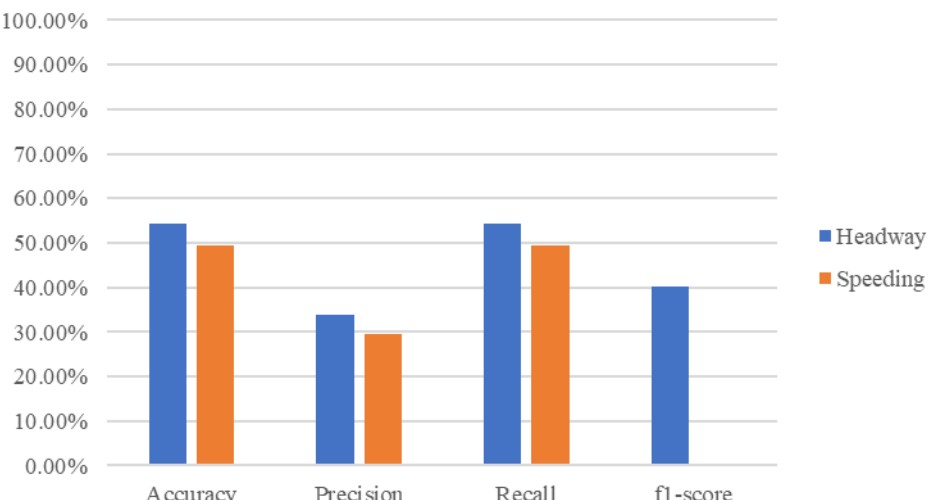

**Figure 13.** Performance of the LSTM model for headway and speeding at a normal level.

A descending trend in both training and validation loss is achieved in the LSTM of UK car drivers for headway (a) and speeding (b) as demonstrated in Figure 14 below. The Divergence or this significant gap between training and validation loss in Figure 14a,b might indicate overfitting (high training performance but poor generalisation).

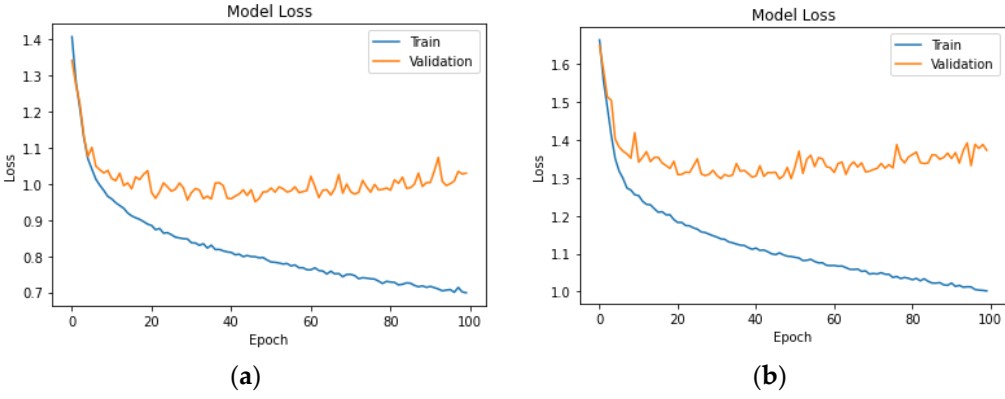

**Figure 14.** Model loss of the LSTM model for UK car drivers for headway (**a**) and speeding (**b**).

## 5. Discussion

The objective of this study was to create and compare machine learning techniques for the detection of risky driving behaviour. The dataset used consisted of trips taken by 30 German drivers, 43 Belgian drivers, and 26 UK drivers. Two machine learning classifiers, LSTM and a neural network, were developed for the analysis.

The effectiveness of the neural network models in predicting headway and speeding levels is encouraging. The high accuracy, precision, and recall rates observed, especially in Germany, demonstrate the potential of these models for real-world applications. Belgium's NN models, while strong, present difficulties in achieving high precision, especially for headway incidents. As for the UK's NN models, the speeding and headway level metrics showcased similar results, with the headway incidents having slightly higher results. The LSTM models in both countries show potential for capturing temporal patterns, but they currently lag behind the NN models in terms of overall accuracy and precision–recall balance. Upon comparing the results of the LSTM model with the earlier neural network models, it is clear that the LSTM model exhibits inferior performance in terms of accuracy,

precision, recall, and F1-score. The LSTM model achieves a test accuracy that falls below the accuracy achieved by the previously mentioned neural network models. Similarly, the precision, recall, and F1-score metrics also indicate poorer performance when compared to the previous models.

Neural network (NN) and long short-term memory (LSTM) models offer distinct advantages in capturing patterns within data. A NN, with its feed-forward structure, excels at discerning intricate relationships, making it effective when temporal dependencies are not prominent. In contrast, LSTM models are designed for time-series data, leveraging their strength in capturing sequential patterns. In our driving behaviour dataset, where temporal dependencies may not heavily influence outcomes, the feed-forward structure of a NN could be more effective in processing contextual information.

The application of LSTM models, despite showing a slightly lower level of accuracy and precision compared to neural networks, presents an intriguing avenue for future research. The ability of LSTM models to capture temporal dependencies and sequential patterns could prove invaluable in predicting. Further optimisation and exploration of LSTM architectures may enhance their performance and reliability in driver behaviour analysis. The smaller dataset size may contribute to the observed inferior performance of the LSTM model. LSTM models often require larger datasets to fully leverage their capacity for learning sequential dependencies. Achieving a balance between model complexity and available data is crucial, and this intricate relationship could be a contributing factor to the observed performance differences.

The models performed well across the board, with each country presenting unique challenges. The variation in road conditions, driving behaviours, and traffic regulations among these countries likely influenced the models' performance disparities. It is crucial to consider these nuances when applying similar predictive models in different geographical contexts. The findings underline the need for continuous refinement and adaptation of machine learning algorithms to address diverse driving environments effectively.

Furthermore, variations in road conditions, infrastructure, and traffic regulations among different countries contribute to divergent driving behaviours. The quality of roadways and adherence to specific rules impact the performance of predictive models, as these models are trained on data that reflect the unique characteristics of each region. Moreover, differences in regulatory environments, including variations in traffic laws and law enforcement practices, can significantly affect driving behaviour. Distinct levels of enforcement for speeding or differing rules regarding headway contribute to the observed disparities in model performance.

In Zou et al. (2022) [34], a controlled experiment was used to predict the drivers' acceleration using both Gated Recurrent Unit (GRU) and Long Short-Term Memory (LSTM), and the experimental results verified the usefulness of the MHMM in personalised driving behaviour analysis and also showed that the performance of GRU is better than that of LSTM. To support the results of the current study, Abdelrahman (2022) [35] presented different machine learning models based on the dataset of a naturalistic driving study. Upon comparison, these models demonstrated the superior performance of the Random Forest Classifier over the Deep Neural Network. Concerning this observation, despite the proven modelling power of DNNs, they seem to show their full potential when dealing with highly non-linear modelling problems with a large number of features and a very large number of training samples (big data). A possible reason why the RF classifier outperformed the DNN in this classification problem may be attributed to the size of the utilised dataset (intermediate size) and the relatively small feature space since only the 14 original features were used to train the DNN.

The results underscore the potential of predictive models for enhancing road safety. However, continuous monitoring, evaluation, and adjustment are essential to ensure their reliability and applicability across diverse driving scenarios.

Numerous noteworthy studies with notable findings are available in this field, as previously mentioned. For instance, Chen et al. (2018) [36] employed a combination of PCA

and multiple linear regression, achieving an impressive accuracy of approximately 97%. Similarly, in [35], the authors utilised the Strategic Highway Research Program 2 (SHRP2) naturalistic driving study (NDS) dataset, comparing various algorithms. They concluded that the Random Forest classifier, with a 10-fold cross-validation, achieved an average accuracy of around 90% and an average F1-score of 0.945. Additionally, Abdelrahman et al. (2019) [37] employed the same SHRP2 NDS dataset, and their proposed RF classifier demonstrated an accuracy of 93.2%, precision of 95.08%, and recall of 93.5%. Furthermore, Chen et al. (2019) [38] conducted a comparative analysis of diverse machine learning algorithms using data acquired through an OBD II decoder in a single car. Their findings, based on the empirical data collected, revealed that the top three ML algorithms were RF, decision tree, and gradient boosting, all achieving validation (test) accuracy above 95%.

In the study by Cura A. et al. (2021) [39], LSTM and CNN-based neural network models were developed to classify and assess bus driver behaviour characterised by deceleration, engine speed pedalling, corner turn, and lane change attempts. CNN architecture indicated better performance indices for the identification of aggressive driving compared to the LSTM network for behavioural modelling, introducing additional results to new models in the field. Furthermore, Parsa et al. (2019) [40] utilised real-time data along with LSTM and GRU (two deep learning techniques) to detect accidents. In the results of this study, the GRU model is observed to perform slightly better than the LSTM model concerning detection rate, showcasing the current models being examined and having enhanced results in the literature in the later years.

The empirical study from Shaanxi Province [41], focusing on distracted driving behaviour using a hybrid neural network (DBRPNN), offers valuable insights complementary to this paper's findings. While our study demonstrates the efficacy of neural networks in predicting risky driving behaviour, the Shaanxi study, with its unique focus on distracted driving, further corroborates the superiority of advanced neural network models. Notably, their DBRPNN model outperformed Bi-LSTM models by 5.42% in accuracy, particularly in short-term predictions. This aligns with our results, where neural networks showed significant predictive accuracy. Both studies underscore the importance of model selection in traffic safety applications and highlight the potential of neural networks in different yet related contexts.

In light of these comparative insights, it becomes clear that, while this study contributes valuable findings, particularly regarding the efficacy of neural networks, the potential of LSTM models in this domain remains an area ripe for future exploration. This aligns with the broader research trajectory, as noted by Philippe Barboda et al. (2023) [33], emphasising continuous innovation and optimisation in machine learning applications for road safety.

Utilising a combination of machine learning algorithms and i-DREAMS data to identify safe driving behaviour holds the promise to transform road safety interventions. Through the utilisation of data-driven insights and advanced analytics, this method has the potential to enhance road safety significantly, leading to a decrease in crashes and, ultimately, preserving lives.

## 6. Conclusions

The outcomes of this study hold significant implications for road safety interventions. Utilising machine learning algorithms and data-driven insights can facilitate the identification of safe driving behaviour, enable prompt feedback to drivers, and foster a safer driving environment. The insights derived from this study play a pivotal role in refining the capabilities of the STZ by providing a deeper understanding of driving behaviour dynamics and improving the prediction of risky driving scenarios. Further research avenues should concentrate on evaluating the long-term effects of interventions, assessing real-time systems, and considering human factors and driver engagement. Additionally, investigating the generalisability and scalability of the developed models and interventions across diverse populations, geographic regions, and vehicle types is vital to ensuring their widespread impact on enhancing road safety.

Acknowledging the study's limitations is crucial. The dataset, drawn from 30 German car drivers, 43 Belgian car drivers, and 26 UK car drivers, may not fully capture the diversity of driving behaviours across various regions and populations and may pose limitations on the generalisability of conclusions on a global scale. While the study provides valuable insights into driving behaviour within the selected regions, it is crucial to acknowledge the potential diversity in driving behaviours across different countries and populations. Moreover, the performance of the LSTM model was comparatively lower than that of the neural network model, suggesting the need for additional optimisation and tuning to enhance outcomes.

Regarding the observed disparities among different countries, it is indeed imperative to address this aspect in the context of future modelling efforts. The varying road conditions, traffic regulations, and driving behaviours across different regions emphasise the importance of considering geographical context-specific attributes. This approach ensures that safety interventions and predictive models are tailored to the unique challenges presented by diverse driving environments.

In summary, this study highlights the potential of machine learning techniques, especially neural networks, in identifying risky driving behaviours and improving road safety. The implementation of real-time applications based on these techniques can offer drivers instant feedback and guidance, enabling them to make informed decisions, enhance their driving habits, and reduce the risk of crashes. While the LSTM model exhibited inferior performance when compared to earlier neural network models, it is important to recognise that each model type has its strengths and limitations. It is really important to acknowledge that no single model may be universally superior for all aspects of driving behaviour prediction.

For future research, a deeper exploration of dataset characteristics, particularly the strength of temporal dependencies, could guide the selection of appropriate models and aim for broader data inclusion to enhance the external validity of the findings and ensure a more comprehensive understanding of global driving behaviour patterns. Investigating the interplay between dataset size, model architecture, and the unique attributes of driving behaviour data is essential for refining model choices. Despite the current performance disparities, the application of LSTM models presents an intriguing avenue for future research, with the potential for further optimisation and exploration of LSTM architectures to enhance their reliability in driver behaviour analysis. Additionally, future research endeavours should focus on integrating contextual information such as weather conditions, road infrastructure, and traffic patterns to enhance the accuracy and applicability of the models. Furthermore, running the same analysis using methods from the updated literature will enhance and strengthen the findings of similar research. Personalised driver modelling, accounting for individual characteristics, can lead to more effective behaviour change interventions. Addressing these areas will advance the comprehension of safe driving behaviour identification, refine intervention systems, and ultimately contribute to enhancing road safety, lowering the occurrence of crashes, and preventing injuries on the roads.

**Author Contributions:** S.R.: software, formal analysis, data curation, writing—original draft preparation; T.G.: software, formal analysis, data curation, writing—original draft preparation; E.M.: conceptualization, methodology, formal analysis, writing—review and editing; T.B.: conceptualization, supervision, resources; G.Y.: conceptualization, supervision, resources. All authors have read and agreed to the published version of the manuscript.

**Funding:** The research was funded by the EU H2020 i-DREAMS project (Project Number: 814761) funded by the European Commission under the MG-2-1-2018 Research and Innovation Action (RIA).

**Institutional Review Board Statement:** Not applicable.

**Informed Consent Statement:** Not applicable.

**Data Availability Statement:** Available upon request.

**Acknowledgments:** This study received funding from the EU H2020 i-DREAMS project (Project Number: 814761).

**Conflicts of Interest:** The authors declare no conflicts of interest.

## Appendix A

Code Snippet for the Neural Network model

```
#Define and compile the neural network model
model = keras.Sequential([
keras.layers.Dense(128, activation='relu', input_shape=(X_train.shape[1],)),
keras.layers.Dense(64, activation='relu'),
keras.layers.Dense(len(label_encoder.classes_), activation='softmax') # Output layer with
appropriate number of classes])
model.compile(optimizer='adam',
              loss='sparse_categorical_crossentropy',
              metrics=['accuracy'])
# Train the model
history = model.fit(X_train, y_train, epochs=100, batch_size=32, validation_split=0.1)
Code Snipper for the LSTM model
#Define the LSTM model with additional details
model_lstm = Sequential()
model_lstm.add(LSTM(128, input_shape=(1, X_train.shape[1]), activation='relu', dropout=0.2,
recurrent_dropout=0.2))
model_lstm.add(LSTM(64, input_shape=(1, X_train.shape[1]), activation='relu', dropout=0.2,
recurrent_dropout=0.2))
model_lstm.add(Dense(len(label_encoder.classes_), activation='softmax'))
# Compile the LSTM model with specific learning rate
optimizer = Adam(learning_rate=0.001)
model_lstm.compile(optimizer=optimizer,
                   loss='sparse_categorical_crossentropy',
                   metrics=['accuracy'])
# Train the LSTM model with a specified batch size
history_lstm = model_lstm.fit(X_train_lstm, y_train, epochs=100, batch_size=64, valida-
tion_split=0.1)
```

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
