# Peer review of "Machine Learning Insights on Driving Behaviour Dynamics among Germany, Belgium, and UK Drivers"

_sustainability, doi:10.3390/su16020518_

Round 1
Reviewer 1 Report
Comments and Suggestions for Authors
The study presents an application of machine learning techniques, specifically Long-Short-Term-Memory Networks (LSTMs) and Neural Networks (NN), to analyze and understand the driving behavior dynamics among drivers from Germany, Belgium, and the UK. The use of the 'Safety Tolerance Zone (STZ)' concept within the i-DREAMS project framework is innovative and could offer significant insights into road safety and driving behavior. The authors have compared the performance of the LSTM and NN models on different metrics and have highlighted the superior performance of the NN models. However, more visual aids like graphs and tables to illustrate these results could enhance clarity and comprehension.
While this paper presents an interesting and innovative approach to driving behavior analysis, several improvements could be made to enhance its quality, completeness, and relevance. Here are some suggestions:
-Update References: The paper would benefit from incorporating more recent references from 2022 and 2023. Including recent literature can ensure that your work is in alignment with the current state of research, and demonstrate that you are addressing gaps or extending the latest in the field.
-Details of Machine Learning Models: It would be helpful to include more details about the machine learning models used in the study. Specifically, the structure of the Long-Short-Term-Memory Networks and Neural Networks (such as the number of layers, types of layers, activation functions, loss functions, optimization algorithms, etc.) should be described more comprehensively. This would give the readers a better understanding of the methods and allow other researchers to reproduce your findings.
-Graphical Representation of Models: Adding a graphical or visualization framework for the machine learning models would enhance understanding and provide a visual aid to comprehend the structure and functionality of the models. This could be in the form of flow diagrams, neural network architecture diagrams, or other suitable visualizations.
-Comparative Analysis: An empirical analysis comparing the performance of your models with other related works would strengthen your findings. This would involve running the same tests or analysis on your dataset using the methods from recent literature and comparing the results with your own. It will give the readers a benchmark to evaluate the efficacy of your models.
-Algorithmic Explanation: If possible, provide a pseudo-code or algorithmic explanation for the machine learning models. This would make the methodology more transparent and allow others to replicate your work more easily.
-Improved Discussion of Results: While the results are clear, a deeper discussion regarding why the Neural Network outperformed the LSTM and what this means for future research could be beneficial. This could include speculation or hypotheses based on knowledge of how these models work.
Comments on the Quality of English LanguageMinor editing of English language required
Reviewer 2 Report
Comments and Suggestions for Authors
This article analyzes the safety levels of participants as they engage in natural driving experiences within the i-DREAMS on-road field trials. To accomplish this objective, the study gathered a series of trips from a sample group consisting of 30 German drivers, 43 Belgium drivers and 26 drivers from United Kingdom. These trips were then input into the machine learning methods to uncover the factors contributing to risky driving behaviour throughout the different stages of the experiment.
1. The connection with the i-DREAMS project should be more clearly defined. From the text of the analysis manuscript it is unclear what the i-DREAMS project has to do with it
2. Based on the text of the article, it is impossible to assess how much you can trust the LSTM network: what parameter settings, how it was trained.
3. Formula (4) on line 240 requires the correct fonts, “𝑃𝑟𝑒𝑐𝑖𝑠𝑖𝑜𝑛”, “𝑅𝑒𝑐𝑎𝑙𝑙” is not italic, the “𝑥” sign must be replaced with multiply.
4. "Two feed-forward multi-layer perceptron models were utilized on a subset" How many layers and how many in each layer?
5. What is the data structure? Why are the results so different for different countries?
6. Table 2. Why F1 - 68.00%? The explanations in the text are unconvincing, since this is not the case in other countries.
7. Table 5, Table 7. Many data are very low for LSTM network. Maybe look at the data itself, what is its peculiarity?
8. Table 8, table 9 may indicate poor tuning of the neural network, and not driver errors?
9. The authors themselves write that “Upon comparing the results of the LSTM model with the earlier neural network models, it is clear that the LSTM model exhibits inferior performance in terms of accuracy, precision, recall, and F1-score.” Why then this article?
10. The authors did not get a result, maybe they should first find an effective solution and then write the text?
I don't think the work is worthy of Sustainability .
Reviewer 3 Report
Comments and Suggestions for Authors
Advantages.
Good practical research in the field of road safety based on LSTM network and neural network approach. A very detailed description of experimental methodology is given. The topic of this article is interesting and meaningful for analysing of driver’s behaviour.
The design of the manuscript is well structured:
- Introduction part is given.
- Literature part is given.
- The methodology part with algorithms is given (materials and methods).
- Experimental results and analysis part is given.
- Discussion and conclusion part is given.
There are no significant criticisms about the research methodology.
Disadvantages:
Is the sample size of your data sufficient to draw global conclusions that driver behavior in other countries will be similar?
Some comments:
- don't found references to Figures 1-6.
- don't found references to Tables 8-10.
Round 2
Reviewer 1 Report
Comments and Suggestions for Authors
Specific Comments and Responses:
Details of Machine Learning Models:
The inclusion of detailed explanations and the code snippet in the appendix is a commendable effort. This certainly helps in clarifying the structure and functionality of the models used and aids in reproducibility.
Response Evaluation: Satisfactory. The detailed methodological descriptions and code provide clarity and allow for replication of the study.
Graphical Representation of Models:
While the rationale for omitting graphical representations is noted, their inclusion could significantly enhance understanding. Given that the authors acknowledge the potential benefits and express a willingness to include these in future work, it would be beneficial if at least a basic form of visualization could be integrated in the current manuscript.
Response Evaluation: Partially satisfactory. The intent to include visual aids in future work is positive, but the current manuscript would still benefit from some form of graphical model representation.
Improved Discussion of Results:
The authors' speculative insights into why the Neural Network may outperform the LSTM is a useful addition. It would be beneficial if this discussion also included references to other studies that have observed similar findings, providing a more robust context for the results.
Response Evaluation: Partially satisfactory. While speculative insights are provided, the discussion could be strengthened with references to other research for a more comprehensive analysis.
Comparative Analysis:
The acknowledgment of the uniqueness of the study and comparison to relevant studies in the discussion provides context for the readers. However, the addition of actual empirical data comparing the proposed models with those in the literature would significantly strengthen the paper.
Response Evaluation: Partially satisfactory. The literature context is helpful, but empirical comparative analysis with current models in the literature would be more convincing.
Algorithmic Explanation:
Providing a code snippet is a positive step towards transparency. To further enhance this, the authors could consider providing a high-level pseudo-code or algorithmic description in the main text, which can be understood even without direct access to the full code.
Response Evaluation: Satisfactory with a suggestion. The code snippet is a good addition, but a high-level algorithmic description in the manuscript could further improve transparency and accessibility.
Moderate editing of English language required
Reviewer 2 Report
Comments and Suggestions for Authors
The revised version is much better and can be accepted.
Author Response
Reviewer #2:
The revised version is much better and can be accepted.
Response: We would like to thank the Reviewer for the valuable recommendations made for the present study!
Round 3
Reviewer 1 Report
Comments and Suggestions for Authors
I will provide a review of the third round based on the response text.
Review 1:
Previous Concern: The need for detailed explanations and code snippets for reproducibility.
Authors' Response: Satisfactory. The authors have provided detailed methodological descriptions and code, enhancing clarity and reproducibility.
Evaluation of Response: The authors have satisfactorily addressed the initial concern by including detailed explanations and code in the appendix.
Review 2:
Previous Concern: The absence of graphical representations of the machine learning models.
Authors' Response: satisfactory. The authors have now included basic graphical representations, which should aid understanding.
Review 3:
Previous Concern: Speculative insights in the discussion section needed strengthening with references to other research.
Authors' Response: Partially satisfactory. The authors have added references to contextualize the discussion, but empirical comparative analysis is still recommended.
Review 4:
Previous Concern: Lack of a high-level pseudo-code or algorithmic description for those without access to the full code.
Authors' Response: Satisfactory with a suggestion. The authors have included a high-level algorithmic description in the main text.
Evaluation of Response: Providing a high-level description in addition to the code snippet is a commendable step that should make the methodology more accessible. It is important to ensure that this description is clear, concise, and understandable to readers who may not have a strong technical background.
Minor editing of English language required
